# History and Development of Clinical Use of Functional Stereotaxy for Radiation Oncologists: From Its Origins to Its Current State

**DOI:** 10.3390/curroncol32120656

**Published:** 2025-11-22

**Authors:** Merrik Goulet, Giuseppina Laura Masucci, Daniel Taussky, Marc Levivier

**Affiliations:** 1Faculté de Médecine, Université de Montréal, Montreal, QC H3T 1J4, Canada; merrik.goulet@umontreal.ca; 2Department of Radiation Oncology, Centre Hospitalier de l’Université de Montréal—CHUM, 1000 Rue St Denis, Montreal, QC H2X 0C1, Canada; 3Centre NeuroKnife, Hôpital de La Tour, 1217 Meyrin, Switzerland; 4Department of Neurosurgery, Faculty of Biology and Medicine, University of Lausanne, 1015 Lausanne, Switzerland

**Keywords:** stereotactic radiosurgery, functional neurosurgery, deep brain stimulation, movement disorders, pain, epilepsy, psychiatric disorders

## Abstract

Stereotactic radiosurgery (SRS) is a precise, noninvasive medical technique that uses focused radiation to treat small targets inside the brain without opening the skull. It was first imagined by the Swedish neurosurgeon Lars Leksell as an alternative to traditional brain surgery for neurological disorders. With advances in imaging and computing, SRS evolved during the 1990s into an essential tool, mainly for treating brain tumors and vascular malformations, and continued to be an alternative to functional neurosurgery. This literature review traces the historical development of stereotactic methods and the pivotal innovations that enabled precise intracranial targeting for SRS, while exploring the treatment evolution and most recent usage of SRS for various neurological disorders such as intractable pain (e.g., trigeminal neuralgia), movement disorders, epilepsy, and psychiatric conditions (e.g., obsessive–compulsive disorder). This review underscores how technological progress and shifting clinical priorities have transformed SRS from a niche neurosurgical technique into a cornerstone of modern clinical practice, with functional SRS representing its latest clinical field of expansion.

## 1. Introduction

Stereotaxy was the original term coined by Horsley and Clarke in 1908 to describe their method of precisely targeting brain structures using a three-dimensional (3D) coordinate system. The term stereotactic emerged later, especially in human neurosurgery, and gained popularity after a consensus at the 1973 meeting of the World Society for Stereotactic and Functional Neurosurgery [1].

Although contemporary applications of stereotactic radiosurgery (SRS) predominantly focus on oncological treatments, its initial purpose was to be an alternative to invasive functional neurosurgery, for example, intractable pain (e.g., trigeminal neuralgia), movement disorders, epilepsy, and psychiatric disorders [2]. The role of functional neurosurgery in the development and implementation of functional SRS has become critical, particularly through a better understanding of underlying brain disorders and better visualization of brain anatomy with precision and minimal invasiveness. Functional radiosurgery has benefited from technological advancements in imaging and treatment delivery systems. This development underscores the need for less invasive neurosurgical interventions that prioritize patient safety and efficacy [3]. Functional radiosurgery was originally based on Gamma Knife technology; it has evolved since its initial application and significantly influences the management of functional brain disorders [4].

This article aims to provide a comprehensive historical overview of the development and clinical applications of functional stereotaxy, from its origin to its present-day context. This review underscores how technological progress and shifting clinical priorities have transformed SRS into a versatile tool with enduring relevance. We believe that knowledge of the history of functional stereotaxy is important for understanding its present-day context and the expansion of radiation oncology. This paper offers radiation oncologists a comprehensive roadmap for tracing the birth, refinement, and modern-day expansion of SRS. It not only illuminates how Leksell’s early vision evolved into today’s Gamma Knife- and Linac-based platforms but also uncovers emerging non-oncologic applications that could redefine their clinical practice.

## 2. Methods

To construct a robust historical and clinical narrative of functional stereotactic neurosurgery and SRS, we conducted an extensive literature review using PubMed and Google Scholar.

For the first portion of our paper regarding the history of stereotaxy and its clinical beginnings, searches were conducted from the beginning of stereotaxy, at the end of the 18th century, up to the development of SRS by Leksell in the 1950s. The term “history” was first combined with “stereotaxy”, and we subsequently added “epilepsy”, “pain”, “movement disorders”, and “psychosurgery”.

For the section on Deep Brain Stimulation (DBS), we used the terms “history” and “deep brain stimulation”, and we combined them with “epilepsy”, “pain”, “movement disorders”, and “psychosurgery”.

Finally, for the section focused on SRS, searches were conducted without date restrictions using the terms “stereotactic radiosurgery” and “functional neurosurgery”, combined with “epilepsy”, “pain”, “movement disorders”, and “psychosurgery” or “psychiatric disorders”.

The authors are fluent in English, French, and German, ensuring a comprehensive assessment of the literature published in these languages.

Peer-reviewed journal articles were studied, such as historical monographs and archival documents, conference proceedings, case reports, and clinical trials. Articles were screened based on title and abstract, and relevant articles were then fully read to assess their inclusion in this study.

Inclusion Criteria: Studies describing the development or clinical application of stereotactic techniques; articles detailing instrumentation, targeting methods, or imaging advancements; reports on outcomes of SRS or DBS in functional indications; and historical accounts of key figures and technological milestones. Exclusion Criteria: Studies focusing on cavernous malformations or cancer treatment, including vestibular schwannomas, as well as gray literature.

Data extraction and organization: The extracted data were organized into a timeline to trace the evolution of stereotactic principles, devices, and clinical applications. The findings were grouped into four major domains of functional neurosurgery. Historical techniques were compared to modern interventions to highlight shifts in clinical practice and ethical considerations. Studies have evaluated the impact of SRS on its present-day context, especially in cases where SRS has emerged as a viable alternative to open surgery. Our analysis combines historical narratives with a comparative evaluation. Limitations Considered: Recognized gaps in the literature account for the underreporting of early radiosurgical outcomes and limited long-term data for functional SRS applications.

## 3. Brief History of Stereotaxy

The history of stereotaxy can be traced back to the 19th century. It is compelling how many different specialists from multiple countries shaped the beginnings of stereotaxy. The idea that different parts of the brain are associated with a specific function was presented by Paul Broca in a meeting in April 1861 [5]. This theory led Wilhelm Dittmar, a German physiologist, to develop a device in 1873, one of the earliest devices used to stabilize and localize intracranial brain structures in animals. His device was created to guide a cutting knife and form lesions in the vasomotor center of the medulla oblongata of rabbits [6]. Dittmar’s work was built on earlier studies initiated by his predecessor Carl Ludwig (1816–1895). Ludwig was one of the founders of modern experimental physiology, and he hypothesized that the contraction of arterial vessels was controlled by a specific region of the brain. Initial surgical approaches to the medulla were performed freehand and lacked precision, often resulting in significant damage. In response, Dittmar engineered a mechanical apparatus to enable more accurate targeting while minimizing harm to the surrounding tissues. This was an early step towards stereotactic principles that would later define the field [6].

Towards the end of the 1800s, numerous cranial localization devices were developed worldwide, mainly to observe and localize cranial and cortical regions in relation to bone landmarks. These included Emil Theodor Kocher’s craniometer in 1892 in Bern, Switzerland, which was the most precise method at the time, compared to similar instruments, such as Wilson’s cyrtometer or the cephalometer of Kroenlein and Köhler [5]. All these devices shared the same goal: to accurately relate the surface structures of the cranium to brain structures in the hopes of realizing the least invasive procedure possible, guiding the creation of an opening through a small trephine hole. Kocher aimed to account for variations in human cranial shapes by developing a more universal apparatus to relate surface topography to brain structure [5]. Around the same period, in 1889, Dmitry Nikolaevich Zernov (1843–1917), a Russian anatomist, presented the encephalometer, a precursor of modern stereotactic apparatuses. This device is based on polar coordinates and can precisely locate any point on the skull or brain surface. It has been used clinically in humans to create burr holes after precisely locating various gyri and fissures in the brain. In 1907, another Russian neurologist and psychologist, Grigory Ivanovich Rossolimo, tried to improve this device, which he called the brain topography [7]. Regrettably, their device did not achieve a significant breakthrough in the English medical literature during this period.

The first stereotactic device was developed in 1905 by Sir Victor Horsley and Robert Henry Clarke, a British surgeon and a physiologist, respectively. In 1908, they introduced their stereotactic apparatus to the public, which utilized Cartesian coordinates to investigate deep brain structures in animals, such as the gray nuclei and thalamus, by referencing bone landmarks [7]. Their goal was not immediately clinical; it was to enable precise, reproducible targeting of deep brain structures for experimental purposes. Their initial experiment, published in 1908, provided evidence that the cerebellum functions as a recipient structure rather than as an efferent one. Although this was the prevailing hypothesis at the time, the use of rudimentary lesion techniques often resulted in extensive damage or complications involving injuries to other parts of the brain, rendering accurate interpretation challenging. Utilizing their precise stereotactic apparatus to accurately locate and conduct targeted electrolysis while preserving the integrity of the cerebellar nuclei, they effectively dispelled any potential doubts [8]. Owing to the precision of their apparatus, they subsequently advanced to the development of brain atlases in cats and monkeys [7]. Their invention was a turning point: it transformed brain surgery from a largely exploratory endeavor into a targeted, science-driven discipline. This has allowed researchers to study the function of specific brain regions by creating lesions or stimulating them with remarkable accuracy. While their initial work focused on basic neuroscience, the long-term clinical aim was to translate this precision into human neurosurgery by mapping the brain in three dimensions and creating atlases.

Prior to this point, although devices for guiding procedures on the human brain were available, no stereotactic devices had been developed. However, in 1918, Aubrey Mussen, a Canadian neuropathologist who worked with Horsley and Clarke, created the first stereotactic device for human use. Unfortunately, he never succeeded in convincing a neurosurgeon to use his apparatus, likely due to the absence of a human brain atlas at the time [9]. Many years passed, and both Mussen’s and the Horsley–Clarke apparatus were forgotten.

In the mid-1940s, key breakthroughs occurred on both sides of the Atlantic. Spiegel (1895–1985) and Wycis (1911–1972) transformed neurosurgery together by bringing precision and structure to brain interventions. Spiegel and Wycis, working in the United States, introduced their modified version of the Horsley–Clarke apparatus for human use in 1947 at Temple University in Philadelphia. In 1946, Swiss physiologist Marcel Monnier established a crucial foundation by linking animal experiments to stereotactic neurosurgery, aspiring to develop a neurosurgical approach that utilizes intracerebral electrodes instead of traditional surgical tools [10]. This led to the first human stereotactic surgery performed in 1947 by Spiegel and Wycis, based on intracranial landmarks to account for extracranial brain variability between humans and with Cartesian coordinates. Their approach used ventriculography for image guidance.

Spiegel and Wycis deployed their new human stereotactic frame across a spectrum of indications (psychiatric illness, pain, cyst aspiration, movement disorders) almost immediately after their 1947 publication in *Science* [11]. Subsequently, they published their own human brain atlas in 1952 [12].

In France, Jean Talairach (1911–2007), a psychiatrist who transitioned to neurosurgery, made significant advancements in the field. In 1947, he developed a specialized stereotactic methodology that utilized a stereotactic frame equipped with a three-coordinate reference system. The following year, in 1948, he performed an inaugural surgical procedure on a 72-year-old male patient. This procedure involved thalamotomy via electrocoagulation targeting the ventromedian and centromedian nuclei to alleviate severe trigeminal neuralgia secondary to herpes zoster ophthalmicus [10]. He continued to refine his methods by introducing ventriculography and intraoperative teleangiographic radiography. To deal with human brain variability caused by the use of external landmarks, he described a standardized anatomical coordinate system in 1952 and published his first atlases in 1957 [10]. Innovations in stereotactic technology and clinical use reached a significant milestone in the 1950s with the introduction of an arc radius stereotactic apparatus developed by the Swedish neurosurgeon Lars Leksell. The initial device for stereotactic radiosurgery (SRS) was not a Gamma Knife but rather an orthovoltage X-ray tube affixed to Leksell’s stereotactic frame. This prototype enabled the targeting of cranial pain pathways, such as the Gasserian ganglion, with submillimetric precision, marking the inception of SRS treatments globally. Subsequently, Leksell investigated the use of high-energy proton beams at Uppsala’s cyclotron, achieving remarkable dose conformity despite encountering logistical and technical obstacles. In pursuit of a more feasible source, Leksell collaborated with Börje Larsson to develop a multi-beam Cobalt-60 unit. The inaugural Gamma Knife prototype was used to treat patients in Stockholm in 1968, becoming the first commercially available SRS system. These developments eventually led to the advancements observed today in linac-based radiosurgery [13,14].

The systems’ utility increased over the years with their integration of a CT scanner, followed by the addition of a dedicated MRI adaptation in 1985 [15]. Figure 1 illustrates the evolution of stereotactic apparatuses and radiosurgery platforms, the major technical milestones in the creation of Leksell’s Gamma Knife, and the eventual use of stereotactic CT- and MRI-guided targeting.

## 4. Clinical Beginnings of Stereotactic Neurosurgery

### 4.1. Treatment of Movement Disorders

As previously mentioned, Wycis and Spiegel were the first to document stereotactic neurosurgery in humans in 1947. However, surgical intervention to treat movement disorders precedes stereotaxic treatment. In 1908, Horsley reported excision of the precentral cortex in a 14-year-old boy with athetoid and clonic movements of the right arm. He used electrodes to locate the area and achieved complete alleviation of spasmodic movements one year later. However, voluntary movement in the left upper limb was almost absent, accompanied by astereognosis, highlighting the need for more precise interventions [16]. Subsequent approaches developed by Putnam in 1938 targeted other structures, such as the pyramidal tract. In 1940, Meyers targeted the caudate nucleus and globus pallidus (GPi). Some improvements were reported, but not full recovery or a lower mortality rate. Subsequent approaches were developed by, for example, Talairach in 1947, who aimed to cure dyskinesia with cortical resection of Brodmann areas 4, 6, and 8 [10]. Most interventions, however, targeted the GP internus with new lesioning methods, such as with the cryogenic probe introduced by Cooper in 1955 [15]. Prior to the advent of DBS, posteroventral pallidotomy was the most common intervention for Parkinson’s disease, effectively addressing tremor, rigidity, and dystonia. Ventrolateral thalamotomy was typically reserved for patients with tremor-dominant presentations. The introduction of L-dopa in 1965 diminished the need for invasive neurosurgical interventions. However, the inability of medication to consistently deliver therapeutic outcomes without adverse effects over time, coupled with advancements in imaging technology, rekindled interest in stereotactic surgery.

### 4.2. Treatment of Epilepsy

Until the late 19th century, trepanation was the sole surgical intervention used in epilepsy treatment. Significant advancements occurred when Victor Horsley pioneered new treatment methods and conducted the first modern epilepsy surgery in 1886 on a 22-year-old male patient. Advancements in technologies such as electroencephalography (EEG), computed tomography (CT), magnetic resonance imaging (MRI), and positron emission tomography (PET) have significantly enhanced our understanding of the mechanisms of epilepsy and localization of activity clusters [17].

### 4.3. Treatment of Psychiatric Conditions

The development of stereotactic frames by Wycis and Spiegel facilitated the exploration of alternative treatment methods, particularly those targeting the thalamus and hypothalamus. This innovation marked the inception of subcortical stereotactic neurosurgical procedures and serves as a foundational model for contemporary psychiatric neurosurgery [12]. Around the same time, Leksell and Talairach pioneered capsulotomy (the lesioning of the anterior limb of the internal capsule) for refractory psychiatric conditions, such as obsessive–compulsive disorders (OCDs) and anxiety disorders. Other targets were explored, such as the hypothalamus, to treat refractory aggressive behavior. Sano attempted this in 1962 and performed posteromedial hypothalamotomy with positive outcomes to reduce such behavior [18]. Additional experiments have been conducted to address various psychiatric disorders with varying degrees of success.

The negative perception of surgical interventions for psychiatric disorders can be attributed to their historical development. The practice of “psychosurgery” was initiated by Swiss psychiatrist Gottlieb Burckhardt, who, in 1888, conducted cerebral cortical excisions on six patients diagnosed with schizophrenia. Without surgical training, he removed parts of the cortex, believing it was the center of the psychiatric disorder. His patients showed improvements but experienced aphasia or seizures, leading to backlash from the medical community [19]. The field reemerged in 1936 when António Egas Moniz, with Almeida Lima, developed frontal lobotomies using a “leucotome,” based on Fulton and Jacobsen’s animal research. Moniz received a Nobel Prize in 1949 for his leukotomies [15,20]. James Watts standardized prefrontal lobotomy in the U.S. and helped with the popularization of Walter Freeman’s transorbital lobotomy technique, which was simpler to perform [21]. In 1949, Fulton proposed cingulotomy, which proved superior to Watts’ procedures and remains in use for OCD, depression, anxiety, and chronic pain [22,23]. However, the introduction of chlorpromazine in 1952 had the same effect as L-dopa in Parkinson’s disease, which caused a significant decrease in interest in lobotomies to treat psychiatric conditions.

Today, neurosurgical approaches for psychiatric conditions are the main focus, including a wide range of targets and techniques, whose indications are better controlled using multidisciplinary approaches.

### 4.4. Treatment of Chronic Pain

The application of stereotactic neurosurgery to chronic pain began in the early 20th century and focused predominantly on trigeminal neuralgia (TN). In the 1930s, Kirschner developed a frame specifically designed to access the foramen ovale for percutaneous injections into the Gasserian ganglion to alleviate facial pain. By the mid-20th century, some stereotactic techniques were also extended to the management of cancer-related pain. In 1953, pituitary gland lesioning was explored as a palliative intervention for metastatic disease [15]. The use of neurosurgery for TN diminished over time after the 1960s due to the introduction of carbamazepine. Interventions targeting the nuclei of the thalamus, especially the ventral posterolateral nucleus, have also been proposed for the treatment of intractable deafferentation pain (anesthesia dolorosa), such as in cases of trauma, amputation, or stroke.

## 5. Deep Brain Stimulation: Treatment Without Destruction

DBS emerged as a direct extension of stereotactic neurosurgical principles, offering a safe, reversible, and ethically acceptable procedure by modulating neural function, rather than permanently lesioning deep brain structures. Enabled by the precision of stereotactic targeting, DBS was initially used intraoperatively to confirm lesion locations. However, in the 1960s and 1970s, some neurosurgeons proposed to also use it as a therapeutic modality. Modern DBS was pioneered in 1987 by neurosurgeon Alim-Louis Benabid and neurologist Pierre Pollack in Grenoble, who published their first paper on the use of thalamic DBS for tremor, with results comparable to those of lesioning while maintaining reversibility [24]. These advances would not have been possible without stereotactic technologies, allowing for safe electrode placement in deep-seated millimeter-scale targets.

### 5.1. Treatment of Movement Disorders by DBS

The most well-established application of DBS is the treatment of movement disorders, particularly Parkinson’s disease, essential tremors, and dystonia. The success of this technique is based on its reliance on stereotactic frames, atlases, and MRI or CT imaging to precisely localize targets. Benabid and Pollak’s landmark use of DBS in the VIM in 1987 to alleviate tremors in Parkinson’s opened the door to broader applications [24]. By the early 1990s, stereotactically guided implantation in the STN and GPi had been validated through animal and clinical trials. These targets remain central to Parkinson’s disease treatment, with STN being better for Parkinsonian symptoms and GPi being chosen often for patients with mood, cognitive, or speech issues. Dystonia, particularly generalized forms, is also treated via GPi stimulation [25,26].

### 5.2. DBS in Epilepsy

From the 1950s to the 1970s, extensive research was conducted to identify precise surgical targets for epilepsy treatment [12]. Key targets identified via stereotactic imaging include the anterior nucleus of the thalamus for limbic seizures, the hippocampus for temporal lobe epilepsy, and the centromedian nucleus for generalized seizures [27]. The pivotal SANTE trial (Stimulation of the Anterior Nucleus of the Thalamus for Epilepsy) in 2010 led to FDA approval of Medtronic’s ANT-DBS system as an adjunctive therapy for adults with drug-resistant partial-onset seizures [28,29]. Advances in imaging, particularly high-resolution MRI, now allow improved direct visualization of deep nuclei, although many procedures still rely on indirect atlas-based targeting [28].

### 5.3. DBS in Psychiatric Disorders

DBS has brought new ethical and technical standards to psychiatric functional neurosurgery by enabling reversible functional modulation without the irreversible damage caused by ablative psychosurgery [18,20,30,31]. These procedures are dependent on precise stereotactic planning, owing to the complexity and variability of the individual limbic anatomy.

### 5.4. DBS in Pain Treatment

Although it is barely used nowadays and lacks FDA approval, DBS for chronic pain has historically been one of its earliest applications. Relying on stereotactic methods, DBS has helped control intractable post-amputation pain, as explored by Mazars in France and Cooper in the United States in the 1970s and 1980s [24,32].

## 6. Radiosurgery and Its Possibilities: The Revival of Its Role in Functional Neurosurgery

The exact mechanism of functional SRS remains debated to this day and is likely multifactorial. SRS allows for the creation of precise intracranial lesions using focused beams of ionizing radiation, thereby minimizing collateral damage. While certain effects are attributed to localized tissue damage, inflammation, and demyelination [33], other studies propose that SRS may induce neuroglial remodeling and functional modulation without significant cellular destruction, or if destruction occurs, it progresses slowly enough to allow for cerebral reorganization [34]. Some researchers propose that it is a combination of both, involving focal necrosis and peripheral neuromodulation effects. Due to the complexity of this subject, it falls beyond the scope of this review and may warrant a dedicated paper. For further exploration, the following references should be consulted with careful consideration [35,36,37].

The first Gamma Knife unit was installed in Stockholm in 1967, but its functional neurosurgical applications remained limited. It was not until the late 1980s that SRS began to gain traction in this field with the development of advanced MRI-guided planning and more precise stereotactic localization. A significant limitation of SRS compared with DBS or open lesioning is its inability to perform intraoperative electrophysiological confirmation [38]. This has led to suboptimal usage of SRS for decades as a treatment option for functional disorders [39]. While Gamma Knife remains the predominant technology in the literature, the use of Linac-based platforms is increasing, especially for trigeminal neuralgia. To our knowledge, SBRT and adaptive image-guided radiotherapy have not yet been reported for functional indications [40].

### 6.1. SRS for Functional Neurosurgery

The following section explores the clinical use of SRS in each of the major domains of functional neurosurgery: movement disorders, epilepsy, psychiatric disorders, and pain syndromes.

### 6.2. Movement Disorders

Although the first Gamma Knife was installed in 1967, only five cases of movement disorders were treated from 1968 to 1970 [39]. The use of SRS for movement disorders expanded in the mid-1980s with improved neuroimaging, better treatment planning systems, and the Gamma Knife developed by Leksell. Preclinical studies on rats and monkeys have helped us explore many parts of the brain, including the caudate nucleus and the ventral intermediate nucleus (VIM), which is still the most used target for Parkinson’s disease. VIM targeting can reduce tremor, bradykinesia, and rigidity to some extent while being safe and precise if used with MRI [41,42]. The International Stereotactic Radiosurgery Society practice guidelines (2019) recommend 130–150 Gy as an effective and well-tolerated dose range [43].

While the Gamma Knife unit still uses a stereotactic frame attached to the patient’s head, frameless SRS techniques, such as CyberKnife and other Linac-based systems, have been investigated and have shown promising early outcomes in terms of safety and efficacy, potentially offering broader access to functional radiosurgical treatment [44,45].

A recent meta-analysis [46] suggested comparable efficacy for Gamma Knife and Linac-based thalamotomy in tremor reduction, but conclusions were limited due to the small number of Linac studies; Gamma Knife was linked to more adverse events and a longer time to improvement.

Table 1 presents some recent studies on tremor with their characteristics and results. The individual studies summarized in this table were selected from the recent systematic reviews and meta-analyses referenced in [47,48].

### 6.3. Epilepsy

The first use of radiation as a potential treatment for epilepsy appeared before the invention of SRS. Talairach performed pioneering work on implanted radioactive seeds in the brain. Initially, he used it in 1954 to treat inoperable tumors with radioactive gold seeds. He then applied, around 1957, yttrium-90 implants in patients with mesial temporal lobe epilepsy (MTLE) and no space-occupying lesions. In his 1974 study, Talairach achieved promising early seizure control rates [12,55,56]. However, other researchers did not replicate these outcomes with comparable success, limiting the popularity of the SRS technique for epilepsy at the time. Functional SRS for epilepsy re-emerged in the mid-1990s through the work of Régis et al. in France, who published encouraging results using Gamma Knife [55].

To evaluate the efficacy of SRS in comparison with traditional open surgery, the ROSE trial was conducted, and its findings were published in 2018. This randomized, single-blinded, controlled trial assessed SRS for anterior temporal lobectomy in patients with pharmacoresistant unilateral MTLE. The results indicated that SRS is a viable alternative to anterior temporal lobectomy (ATL) for patients who have contraindications to or are hesitant to undergo open surgery [57]. A 2025 systematic review and meta-analysis comparing laser interstitial thermal therapy, radiofrequency ablation, and stereotactic radiosurgery for mesial temporal lobe epilepsy provides contemporary comparative outcome data and helps contextualize the ROSE trial results within evolving minimally invasive options [58]. Authors explain that the three treatment methods are comparable, even if laser therapy seems more promising. However, there is no doubt that more RCTs for SRS are necessary in the future.

Table 2 summarizes recent mesial temporal lobe epilepsy studies with key characteristics and results.

In addition to MTLE, SRS has also been applied to treat hypothalamic hamartomas, which are difficult to access in open surgery. SRS has shown good results in reducing seizure frequency, but without seizure freedom, and offers a better risk–benefit ratio than surgical methods [55,62].

### 6.4. Psychiatric Conditions

The use of stereotactic radiosurgery for psychiatric disorders began with Lars Leksell in 1953, when he treated a patient with intractable obsessive–compulsive disorder (OCD) using a 300-kV X-ray apparatus [34,63]. The patient was a 29-year-old man with intractable OCD. This stereotactic radiosurgery operation was performed in two stages: First, an anterior gamma capsulotomy on the right side was performed with a collimator helmet with an 8 mm diameter and a maximum dose of 100 Gy to the most anterior part of the internal capsule, guided via pneumoencephalography. One month later, the same procedure was performed with a maximum dose of 120 Gy on the left side. No information was reported regarding the follow-up of this patient [64]. Similarly to other functional treatments, the application of SRS was constrained prior to the development of advanced imaging techniques. It was not until 1978 that Rylander published an inaugural outcome study following the Gamma Knife capsulotomy conducted by the Karolinska group. This study involved nine patients, five of whom were treated for OCD, yielding reasonable results [65]. In the following decades, Gamma Knife surgery for OCD was performed in Sweden, but due to a lack of radiobiology knowledge at the time, it led to excessively high doses, resulting in many adverse effects, which limited the clinical use of SRS for many years [66].

Current radiosurgical approaches most commonly target the anterior portion of the internal capsule; outcome data indicate efficacy comparable to alternative anterior capsulotomy techniques [67]. Table 3 provides an overview of recent series, reporting patient numbers, dosing, targeting, efficacy, and adverse events [68,69].

### 6.5. Pain

For pain management, neurosurgery and later SRS were investigated for the treatment of two main entities: trigeminal neuralgia (TN) and intractable pain. In 1951, Leksell performed the first radiosurgical procedure for TN targeting the Gasserian ganglion. Like other SRS procedures, interest in the treatment of TN increased in the early 1990s with the availability of MRI, which enabled improved direct visualization of the nerve [75]. With time, the target moved closer and closer to the nerve root. In 1993, Rand targeted the cisternal segment of the nerve, while Lindquist et al. targeted the nerve at its emergence from the brainstem, the trigeminal nerve root entry zone (REZ), which is now the most used target. Doses of 70–90 Gy were demonstrated to be safe and effective in 1996 in the multicenter trial conducted by Kondziolka and a prospective controlled trial in 2006 [34,75]. However, microvascular surgeries remain the gold standard treatment for TN, proven to have lower rates of short-term and long-term pain freedom than SRS [76].

Recent studies report comparable clinical outcomes between Gamma Knife and frameless Linac-based or CyberKnife systems for trigeminal neuralgia [77,78,79], although meta-analyses continue to favor conventional procedures such as microvascular decompression or rhizotomy [80,81]. Table 4 summarizes recent series with treatment details and outcomes.

The first SRS thalamotomies for intractable pain were performed by Leksell in 1968 using a Gamma Knife, which showed, as opposed to other functional disorders, an immediate effect in approximately two-thirds of the patients. This rapid pain reduction was likely due to the high doses of radiation, sometimes exceeding 250 Gy, which caused severe side effects. In 1972, Backlund first attempted pituitary SRS for the treatment of malignant pain [89]. In the 1980s, research on ventromedial thalamotomy was explored in cancer patients, using SRS as a pain-relieving method, but the effect was variable and not maintained for a sufficient amount of time [90].

### 6.6. Linking Historical Evolution to Current Radiotherapy Paradigms

The historical evolution of stereotactic radiosurgery, from early lesioning and frame-based stereotaxis to modern image-guided, frameless systems, directly informs its present-day context. Innovations in target localization, dose conformality, and motion management that originated in functional radiosurgery have been translated into SBRT and adaptive image-guided radiotherapy, enabling higher ablative doses, hypofractionation, and on-table plan adaptation while preserving precision and normal-tissue sparing. Explicitly linking these developments clarifies how past technical and conceptual advances underpin current clinical capabilities and helps clinicians appreciate the rationale for adopting newer platforms and workflows. Recent advances in other radiation-based disciplines illustrate how integrated, personalized care can reshape therapeutic paradigms, for example, total neoadjuvant therapy [91], organ-preservation strategies [92], and AI-driven treatment personalization in rectal cancer [93]. By analogy, stereotactic radiosurgery is positioned to follow a similar trajectory toward greater individualization and function preservation through combined modality strategies [94], refined patient selection supported by radiomics and predictive analytics [95], adaptive image-guided planning including MR-guided adaptive radiotherapy techniques [96], and the incorporation of predictive analytics and machine learning to optimize dose, fractionation, and target selection for each patient [97].

## 7. Ethical Challenges

Stereotactic radiosurgery for functional brain disorders carries unique ethical challenges that echo the history of ablative psychosurgery. Unlike Deep Brain Stimulation, SRS effects are irreversible once the radiation induces tissue damage. Therefore, patients must be counseled extensively on both the potential benefits and permanence of their treatment. The historical context of stereotaxy emphasizes the need to acknowledge the legacy of psychosurgery abuse in building trust and avoiding repeat missteps. Stereotactic professional societies should continue to champion consensus guidelines for target selection and radiation dosing to reduce practice variability and enhance patient safety.

## 8. Conclusions

Stereotaxy has undergone remarkable evolution since the late 19th century, driven by innovations in stereotactic targeting and imaging technologies. It epitomizes the culmination of over a century of advancements in stereotactic and radio- and neurosurgery, achieved through the contributions of numerous researchers from various countries. Stereotactic radiosurgery (SRS) has evolved significantly since its inception in 1951 as a noninvasive alternative to functional neurosurgery. Although initially limited by imaging constraints, advances in MRI-guided planning have expanded SRS applications beyond tumors to functional disorders, such as tremor, epilepsy, OCD, and trigeminal neuralgia.

SRS is a noninvasive option for carefully selected patients who have failed medical therapy or cannot undergo surgery, delivered with precise, image-guided stereotactic targeting. Multidisciplinary collaboration between neurosurgeons and neurologists is essential. As imaging and delivery systems continue to advance, radiation oncologists should remain aware of the emerging functional applications of SRS. These innovations may expand treatment options for patients with neurological and psychiatric disorders.

Because the current literature is largely retrospective, high-quality prospective studies and randomized trials are necessary to establish whether SRS offers meaningful advantages over standard treatments.

## Figures and Tables

**Figure 1 curroncol-32-00656-f001:**
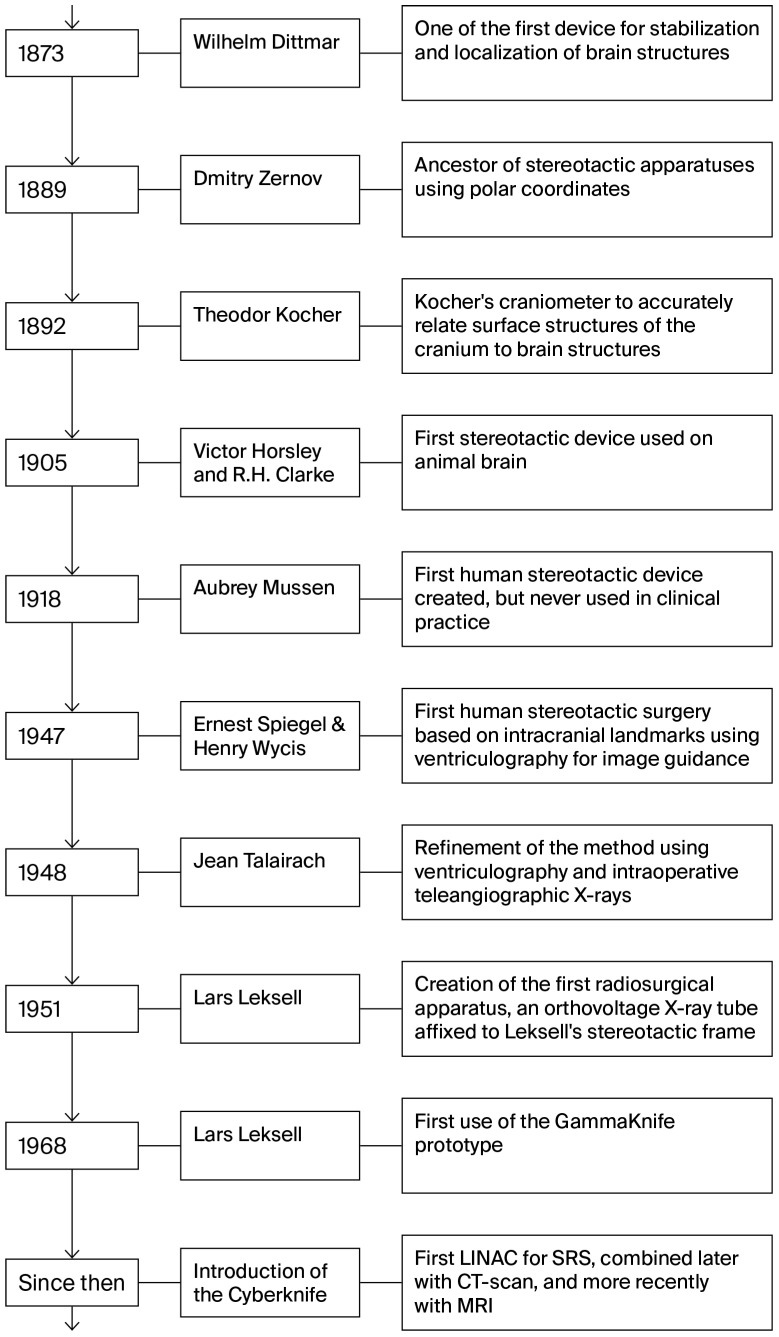
Evolution of stereotactic apparatuses and radiosurgery platforms. This timeline maps the major technical milestones from the first polar coordinate devices of the late 19th century through the advent of human stereotactic frames to the creation of Leksell’s Gamma Knife and the integration of stereotactic CT- and eventual MRI-guided targeting.

**Table 1 curroncol-32-00656-t001:** Summary of studies evaluating reduction in tremor in patients with essential tremors and/or Parkinson’s disease with stereotactic radiosurgery (SRS) treatment. RR, response rate—the maximal response in the cohort at any point; RR 12 m, response rate at 12 months of follow-up; AE, adverse effect.

Author and Year	Patients	Radiotherapy Modality	Maximal Dose	Target	Efficacy	Toxicity
Niranjan, 2017 [41]	73	Gamma Knife	130–150 Gy	VIM	RR: 93.2% RR 12 m: N/A	AE: 3.8% Hemiparesis; facial weakness; dysphasia; numbness in the contralateral hand
Pérez-Sanchez, 2020 [49]	13	Gamma Knife	130 Gy	VIM	RR: 84.6% RR 12 m: 63.6%	AE: 23% Paraesthesia; minor cognitive complaints; depression
Khattab, 2021 [45]	33	Gamma Knife	160 Gy	VIM	RR: 83% RR 12 m: 50%	AE: 6% Headache
Ochiai, 2021 [50]	17	Gamma Knife	130 Gy	VIM	More than 50% decreased tremor: RR: 77% RR 12 m: 71%	AE: 12% Motor weakness; neurological deficit
Luo, 2022 [51]	23	LINAC	145–160 Gy	VIM	RR: 82.6% RR 12 m: N/A	AE: N/A
Ankrah, 2023 [52]	42	LINAC	135 Gy	VIM	RR: 89.7% RR 12 m: N/A	AE: 2.4%, severe; 9.5%, mild
Horisawa, 2023 [53]	27	Gamma Knife	130 Gy	VIM	RR: 88.9% RR 12 m: N/A	AE: 22% Complete hemiparesis; foot weakness; dysarthria; dysphagia (death by pneumonia); lip and finger numbness
Tuleasca, 2023 [54]	78	Gamma Knife	130 Gy	VIM	RR: 67.6% RR 12 m: N/A	AE: 8.9% Transient hemiparesis

**Table 2 curroncol-32-00656-t002:** Summary of studies evaluating reduction in seizures in patients with MTLEs with SRS treatment. RR, response rate—the maximal response in the cohort at any point; AE, adverse effect.

Author and Year	Patients	Radiotherapy Modality	Dose	Target	Efficacy	Toxicity
Usami, 2012 [59]	7	Gamma Knife	18–25 Gy at 50% isodose	Amygdala, hippocampal head and body, most of the parahippocampal gyrus, and the entorhinal cortex	29% seizure remission	29% symptomatic radiation necrosis (headache, edema, gait disturbance); 14% death before seizure control
Kawamura, 2012 [60]	11	Gamma Knife	20–25 Gy at 50% isodose	Anterior 2.5 cm of the hippocampus, the amygdala, and parahippocampal gyrus	36% seizure remission	9% death due to seizure; 9% cognitive impairment, aphasia, and hemiparesis; 9% severe headache and visual changes; 9% cognitive impairment and hemiparesis
Wang, 2017 [61]	37	Gamma Knife	15–25 Gy at 50% isodose	Edges of amygdaloid nucleus and hippocampal area and edges of forehead and anterior temporal lobe	Seizure-free N/A RR: 89.2%	8% mental symptoms; 3% extradural hematoma; 3% memory decline
Barbaro, 2018 [57]	31	Gamma Knife	24 Gy at 50% isodose	Amygdala, anterior 2 cm of hippocampus and parahippocampal gyrus	52% seizure remission	AE: 45.2% Headaches, transient neurological deficits, transient exacerbation of seizures; cerebral edema; pin site infection

**Table 3 curroncol-32-00656-t003:** Summary of studies evaluating reduction in symptoms in patients with OCD with SRS treatment. FR, full response described in the literature as a greater than 35% improvement in the Y-BOCS score (* greater than or equal to a 25% improvement); Remission, defined as an endpoint Y-BOCS total score ≤ 12 or 8; VP, ventral portion; ALIC, anterior limb of the internal capsule.

Author and Year	Patients	Radiotherapy Modality	Maximal Dose	Number of Shots	Target	Efficacy	Toxicity
Rasmussen, 2018 [70]	55	Gamma Knife	180 Gy	Single: 2 Single repeated: 13 Double: 40	VP of ALIC bilat	FR: 56% (FR of 7% with single shot vs. double shots)	9% transient edema; 5% cyst; 1.8% radio-necrosis; transient headache
Gupta, 2019 [71]	40	Gamma Knife	120–180 Gy	N/A	VP of ALIC bilat	FR: 45% Remission: 40%	25% mood disturbance; 7.5% neurological complications (headache with difficulty in speech, dizziness, tinnitus, forgetfulness); 2.5% radio-necrosis; 20% weight changes
Peker, 2020 [72]	21	Gamma Knife	140–150 Gy	Single: 1 Double: 20	VP of ALIC bilat	FR: 75% Remission: 35%	23.8% headache (14.3% transient, 9.5% persistent); 9.5% cyst
Ertek, 2021 [73]	12	Gamma Knife	140–180 Gy	Single: 3 Double: 9	ALIC bilat	FR: 50% Remission: 16.7%	16.7% headache
Pattankar, 2022 [74]	9	Gamma Knife	120–160 Gy	Double: 3 Triple: 5 Nonuple: 1	1 anterior portion of the cingula; 8 midputaminal points of ALIC bilat	FR *: 44.4%	None

**Table 4 curroncol-32-00656-t004:** Summary of studies evaluating reduction in pain in patients with TN with SRS treatment. Pain-free—Grade I to IIIa on the Barrow Neurological Institute scale (* also included Grade IIIb); RG, retrogasserian target on the trigeminal nerve; REZ, nerve root entry zone.

Author and Year	Patients	Radiotherapy Modality	Maximal Dose	Target	Efficacy	Toxicity
Debono, 2019 [82]	301	LINAC	90 Gy	RG	Pain-free: 82% at 3 months Recurrence: 26.4%	26.2% facial hypesthesia; 19.6% paresthesia; 1.3% eye irritation
Koca, 2019 [83]	21	LINAC	70 Gy	Trigeminal Cistern	Pain-free: 57.1% at 16 months At least once for everyday pain: 90.5% Recurrence: not significant	52.4% hypoesthesia; dysphagia and paresthesia 61.9%; difficulties eating 57.1%; difficulties speaking 52.4%; 33.3% vision impairment
Barzaghi, 2020 [84]	112	Gamma Knife	70–90 Gy	RG	Pain-free: 84.8% at 6 months Recurrence: 72%	14.1% hypoesthesia
Kienzler, 2021 [85]	234	LINAC	90 Gy	REZ	Maximal pain-free *: 91% and 88.1% at 12 months Recurrence: 27.8%	32.4% hypesthesia; 4.7% dry eye syndrome; 0.4% anesthesia dolorosa; 0.4% hearing loss
Okunlola, 2023 [86]	153	Gamma Knife	70–80 Gy	Trigeminal Cistern	Pain-free: 78.3% Recurrence: 9.2%	1.3% transient headache, giddiness, and imbalance; 3.3% facial numbness
Orlev, 2023 [87]	162	Gamma Knife	70–90 Gy	REZ	Pain-free *: 80% at 12 months Recurrence: N/A	N/A
Sato, 2023 [88]	103	Gamma Knife	90 Gy	RG	Pain-free: 82.5% at initial evaluation and 58.2% at 10 years Recurrence: 27.1%	24.3% facial numbness
Shields, 2024 [77]	116	LINAC	80–90 Gy	REZ: 31.9% Meckel’s cave: 68.1%	Pain-free *: 94.8% at last follow-up Recurrence: N/A	36.2% fatigue; 4.3% pain flare; 1.7% dry eye

## Data Availability

No new data were created or analyzed in this study.

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
