# Peer review of "History and Development of Clinical Use of Functional Stereotaxy for Radiation Oncologists: From Its Origins to Its Current State"

_curroncol, 2025, doi:10.3390/curroncol32120656_

Round 1
Reviewer 1 Report
Comments and Suggestions for Authors
The authors described history and current application of functional stereotaxy. This is a good overview. Since the title is aimed to radiation oncologist's readership, it is recommended that the authors provide radiotherapeutic and technical details such as dose prescription used for various functional disorders, description of target localization and delineation, the outcomes of success via more detailed literature review for each functional disorders and the limitation of its use as well as potential/future direction. These details will provide more comprehensive summary to the readers.
Comments on the Quality of English LanguageNo issues.
Author Response
The authors described history and current application of functional stereotaxy. This is a good overview. Since the title is aimed to radiation oncologist's readership, it is recommended that the authors provide radiotherapeutic and technical details such as dose prescription used for various functional disorders, description of target localization and delineation, the outcomes of success via more detailed literature review for each functional disorders and the limitation of its use as well as potential/future direction. These details will provide more comprehensive summary to the readers.
We thank the reviewer for this important recommendation and agree that radiation‑specific details improve clinical utility for a radiation oncology audience. To address this, we have ensured that dose prescriptions, target localization and delineation, reported efficacy and toxicity, and key limitations are presented in tabular form for the four main indications (Tables 1–4).
Reviewer 2 Report
Comments and Suggestions for Authors
The paper provides an interesting historical overview of the use of radiosurgery in treating non-cancerous diseases.
However, its value as a scientific work in an oncology journal is reduced by a number of shortcomings.
In the 'Materials and Methods' section, a schematic presentation of the data collection process should be considered, including the inclusion and exclusion criteria.
While the data in the subsequent chapters are interesting, they lack detail. There is no information on the dose, radiation regimen, efficacy or toxicity of the treatment.
This information should be presented in tabular form.
In their conclusions, the authors encourage wider use of SRS, yet the paper does not explain why. It is unclear whether this treatment provides any benefit. Such conclusions can only be drawn once the effectiveness of this treatment has been established.
Author Response
The paper provides an interesting historical overview of the use of radiosurgery in treating non-cancerous diseases.
However, its value as a scientific work in an oncology journal is reduced by a number of shortcomings.
In the 'Materials and Methods' section, a schematic presentation of the data collection process should be considered, including the inclusion and exclusion criteria.
We appreciate the request for greater methodological transparency and have added a detailed, reproducible description of how historical sources were identified, selected, and categorized.
While the data in the subsequent chapters are interesting, they lack detail. There is no information on the dose, radiation regimen, efficacy or toxicity of the treatment.
This information should be presented in tabular form.
We thank the reviewer for this observation. In response, we have added detailed, study‑level information on dose, fractionation/regimen, target localization/delineation, efficacy, and toxicity for the four main indications (tremor, mesial temporal lobe epilepsy, obsessive‑compulsive disorder, and trigeminal neuralgia). These data are presented in Tables 1–4.
In their conclusions, the authors encourage wider use of SRS, yet the paper does not explain why. It is unclear whether this treatment provides any benefit. Such conclusions can only be drawn once the effectiveness of this treatment has been established.
We thank the reviewer for this important point. We have tempered our conclusion to reflect current evidence limitations and clarified that broader adoption of SRS should follow confirmation of effectiveness from prospective, comparative studies. Specifically, we replaced the original recommendation with a nuanced statement that emphasizes conditional support for wider use only when supported by high‑quality data.
Reviewer 3 Report
Comments and Suggestions for Authors
This manuscript provides a comprehensive historical overview of stereotactic techniques, tracing their evolution from early neurosurgical tools to modern functional radiosurgery. It highlights the clinical expansion of SRS across movement disorders, epilepsy, psychiatric conditions, and pain management, while suggesting its growing relevance in contemporary radiation oncology.
Strengths
1)Provides a comprehensive and well-structured historical narrative of stereotaxy, from experimental origins to its clinical applications in functional neurosurgery.
2) The chronological progression is coherent and supported by relevant historical figures and technological milestones, enhancing readability.
3) The inclusion of multiple clinical domains (movement disorders, epilepsy, psychiatric conditions, pain) offers valuable breadth for radiation oncologists seeking context beyond oncology.
4) The conclusion successfully transitions toward the future of SRS, emphasizing its expanding role and clinical relevance.
What I recommend improving:
1) The Methods section is overly general; specifying how historical sources were selected or categorized would improve methodological transparency.
2) Certain historical segments are overly detailed (e.g., 19th-century apparatus descriptions) and could be slightly condensed to maintain clinical focus for modern readers.
3) The manuscript would benefit from linking historical evolution to current radiotherapy paradigms, such as SBRT or adaptive image-guided techniques, to reinforce relevance for today's practitioners.
4) The clinical relevance of the manuscript could be improved. To strengthen the forward-looking perspective, a comparison with evolving trends in other radiation-based fields could be insightful. This is an (optional) example that could enhance the quality: recent literature highlights how total neoadjuvant therapy, organ preservation strategies, and AI-driven personalization are reshaping therapeutic paradigms in rectal cancer (such as this review https://doi.org/10.3390/jcm14030912). Introducing this parallel would underscore how SRS may similarly evolve toward individualized, function-preserving care.
Overall it's a good review that is worth publishing after the minor revision that I suggested.
Author Response
This manuscript provides a comprehensive historical overview of stereotactic techniques, tracing their evolution from early neurosurgical tools to modern functional radiosurgery. It highlights the clinical expansion of SRS across movement disorders, epilepsy, psychiatric conditions, and pain management, while suggesting its growing relevance in contemporary radiation oncology.
Strengths
1)Provides a comprehensive and well-structured historical narrative of stereotaxy, from experimental origins to its clinical applications in functional neurosurgery.
2) The chronological progression is coherent and supported by relevant historical figures and technological milestones, enhancing readability.
3) The inclusion of multiple clinical domains (movement disorders, epilepsy, psychiatric conditions, pain) offers valuable breadth for radiation oncologists seeking context beyond oncology.
4) The conclusion successfully transitions toward the future of SRS, emphasizing its expanding role and clinical relevance.
We thank the reviewer for their positive assessment and encouragement. The reviewer’s points highlight the manuscript’s value and helped sharpen our revisions.
What I recommend improving:
1) The Methods section is overly general; specifying how historical sources were selected or categorized would improve methodological transparency.
We appreciate the request for greater methodological transparency and have added a detailed, reproducible description of how historical sources were identified, selected, and categorized.
2) Certain historical segments are overly detailed (e.g., 19th-century apparatus descriptions) and could be slightly condensed to maintain clinical focus for modern readers.
We appreciate the reviewer’s suggestion to condense the 19th‑century apparatus descriptions. However, this review’s explicit aim is to document the full historical lineage that led to modern functional stereotaxy and to show how incremental mechanical and conceptual advances established the principles that underlie today’s image‑guided SRS. We believe thattThe early apparatuses (Dittmar, Kocher, Zernov, Horsley–Clarke) are essential milestones and help us better appreciate the present innovations.
3) The manuscript would benefit from linking historical evolution to current radiotherapy paradigms, such as SBRT or adaptive image-guided techniques, to reinforce relevance for today's practitioners.
We thank the reviewer for this suggestion. We have added a short paragraph linking the historical development of stereotactic radiosurgery to contemporary paradigms such as SBRT and adaptive image‑guided radiotherapy at the end of the Discussion under “7.0.” titled “7.0 Linking historical evolution to current radiotherapy paradigms”:
The historical evolution of stereotactic radiosurgery, from early lesioning and frame‑based stereotaxis to modern image‑guided, frameless systems, directly informed contemporary radiotherapy paradigms. Innovations in target localization, dose conformality, and motion management that originated in functional radiosurgery have been translated into SBRT and adaptive image‑guided radiotherapy, enabling higher ablative doses, hypofractionation, and on‑table plan adaptation while preserving precision and normal‑tissue sparing. Explicitly linking these developments clarifies how past technical and conceptual advances underpin current clinical capabilities and helps clinicians appreciate the rationale for adopting newer platforms and workflows.
4) The clinical relevance of the manuscript could be improved. To strengthen the forward-looking perspective, a comparison with evolving trends in other radiation-based fields could be insightful. This is an (optional) example that could enhance the quality: recent literature highlights how total neoadjuvant therapy, organ preservation strategies, and AI-driven personalization are reshaping therapeutic paradigms in rectal cancer (such as this review https://doi.org/10.3390/jcm14030912). Introducing this parallel would underscore how SRS may similarly evolve toward individualized, function-preserving care.
We thank the reviewer for this constructive suggestion. We have added a short comparative paragraph linking contemporary trends in other radiation‑based fields to potential future directions for SRS, emphasizing individualized, function‑preserving approaches at the end of the Discussion titled “7.0 Linking historical evolution to current radiotherapy paradigms
“Recent advances in other radiation‑based disciplines illustrate how integrated, personalized care can reshape therapeutic paradigms; examples include total neoadjuvant therapy, organ‑preservation strategies, and AI‑driven treatment personalization in rectal cancer. By analogy, stereotactic radiosurgery is positioned to follow a similar trajectory toward greater individualization and function preservation through combined modality strategies, refined patient selection, adaptive image‑guided planning, and incorporation of predictive analytics to optimize dose, fractionation, and target selection for each patient.”
Round 2
Reviewer 2 Report
Comments and Suggestions for Authors The authors have done an excellent job. I recommend accept and publish the manuscript in its current form.Author Response
The authors have done an excellent job. I recommend accept and publish the manuscript in its current form.
Thank you very much for your nice comment
Reviewer 3 Report
Comments and Suggestions for Authors
Most of my concerns have been handled adequately.
Please make sure you cite any bibliographic references in the last (new) paragraph accordingly:
“Recent advances in other radiation‑based disciplines illustrate how integrated, personalized care can reshape therapeutic paradigms; examples include total neoadjuvant therapy, organ‑preservation strategies, and AI‑driven treatment personalization in rectal cancer. By analogy, stereotactic radiosurgery is positioned to follow a similar trajectory toward greater individualization and function preservation through combined modality strategies, refined patient selection, adaptive image‑guided planning, and incorporation of predictive analytics to optimize dose, fractionation, and target selection for each patient.”
Author Response
Most of my concerns have been handled adequately.
Please make sure you cite any bibliographic references in the last (new) paragraph accordingly:
“Recent advances in other radiation‑based disciplines illustrate how integrated, personalized care can reshape therapeutic paradigms; examples include total neoadjuvant therapy, organ‑preservation strategies, and AI‑driven treatment personalization in rectal cancer. By analogy, stereotactic radiosurgery is positioned to follow a similar trajectory toward greater individualization and function preservation through combined modality strategies, refined patient selection, adaptive image‑guided planning, and incorporation of predictive analytics to optimize dose, fractionation, and target selection for each patient.”
We appreciate your constructive comments. We have carefully reviewed the final paragraph and ensured that all bibliographic references are cited appropriately, in accordance with the journal’s guidelines. Please let us know if any further adjustments are needed.
Round 3
Reviewer 3 Report
Comments and Suggestions for Authors
Ok.